# Free Radicals Mediated Redox Signaling in Plant Stress Tolerance

**DOI:** 10.3390/life13010204

**Published:** 2023-01-10

**Authors:** Krishna Kumar Rai, Prashant Kaushik

**Affiliations:** 1Centre of Advance Study in Botany, Department of Botany, Institute of Science, Banaras Hindu University (BHU), Varanasi 221005, Uttar Pradesh, India; 2Instituto de Conservación y Mejora de la Agrodiversidad Valenciana, Universitat Politècnica de València, 46022 Valencia, Spain

**Keywords:** ROS, RNS, RSS, signaling, post-translational modification, stress tolerance

## Abstract

Abiotic and biotic stresses negatively affect plant cellular and biological processes, limiting their growth and productivity. Plants respond to these environmental cues and biotrophic attackers by activating intricate metabolic-molecular signaling networks precisely and coordinately. One of the initial signaling networks activated is involved in the generation of reactive oxygen species (ROS), reactive nitrogen species (RNS), and reactive sulfur species (RSS). Recent research has exemplified that ROS below the threshold level can stimulate plant survival by modulating redox homeostasis and regulating various genes of the stress defense pathway. In contrast, RNS regulates the stress tolerance potential of crop plants by modulating post-translation modification processes, such as *S*-nitrosation and tyrosine nitration, improving the stability of protein and DNA and activating the expression of downstream stress-responsive genes. RSS has recently emerged as a new warrior in combating plant stress-induced oxidative damage by modulating various physiological and stress-related processes. Several recent findings have corroborated the existence of intertwined signaling of ROS/RNS/RSS, playing a substantial role in crop stress management. However, the molecular mechanisms underlying their remarkable effect are still unknown. This review comprehensively describes recent ROS/RNS/RSS biology advancements and how they can modulate cell signaling and gene regulation for abiotic stress management in crop plants. Further, the review summarizes the latest information on how these ROS/RNS/RSS signaling interacts with other plant growth regulators and modulates essential plant functions, particularly photosynthesis, cell growth, and apoptosis.

## 1. Introduction

In the 21st century agriculture and various climatic stresses, such as high temperature, drought, and salinity, have redundantly affected crop growth and productivity, prompting severe threats to global food security for ever-growing global populations [1]. In the Asian continent, a rainfed agriculture system is usually standard and is followed by most farmers. These climatic stresses have become a daunting challenge that has imposed severe repercussions on crop health, thereby ultimately affecting its productivity to a certain extent and leading to livestock death [2]. In addition to sessile, plants are constantly exposed to these climate extremes that instigate various cellular, physiological, biochemical, and molecular responses. Reactive oxygen species (ROS) and reactive nitrogen species (RNS) act as integral components of signal transduction processes regulating vital functions in plants exposed to climate extremes [3]. Initially, ROS and RNS are considered toxic molecules where elevated levels provoke oxidative stress in plants leading to cellular damage and death [4]. However, several recent reports have highlighted that ROS and RNS also function as signaling molecules (when their generation is critically maintained below a threshold level by antioxidative systems), catalyzing several oxidation reactions, thereby modulating vital signaling cascades [5]. 

Redox chemistry is inextricably engaged in the generation, regulation, and sustainment of life on earth by exhilarating reduction–oxidation (redox) reactions essential for driving crucial cellular and metabolic processes such as photosynthesis, respiration, and other biochemical reactions in diverse life forms [6]. Interestingly, recent reports have restricted the involvement of ROS and RNS in stimulating thermodynamically favorable reactions that are essential for sustaining life, including (i) their ability to enhance metabolic reactions, (ii) regulate enzymatic and non-enzymatic antioxidants, (iii) cross barriers (membranes) to activate signaling cascades, and (iv) provide a source of energy (electrons) to defend against oxidative stress [7]. In the recent decade, a plethora of research has been conducted to assess the positive side of ROS/RNS signaling in plants’ growth, development, and defense response. A concomitantly large body of literature has pinpointed their exemplary role [8]. 

Reactive sulfur species (RSS) is a term still to be entered into the general scientific vocabulary due to its low expression and lack of consideration of its role in signal transduction [9,10]. Early prebiotic life forms, i.e., before photosynthesis, likely thrived under a sulfur-rich environment, and several reports claimed that when life originated, approx. 3.8 billion years ago, RSS was the first reactive molecule that influenced expansion [9]. Before photosynthesis, RSS, mainly hydrogen sulfide (H_2_S) released from volcanic eruptions and other geothermal activity, served as building blocks for nucleic acid biosynthesis and protein synthesis for early life forms, such as Beggiatoa, pupfish, giant tubeworms, and mollusks [10]. Researchers have also confirmed the involvement of RSS in providing reducing powers for fixing CO_2_ via the Calvin cycle in various green and purple sulfur bacteria [6,11]. Furthermore, a large body of literature has implicated the significant role of RSS in initiating oxidation reactions, thereby controlling redox homeostasis, cell signaling, and defense response in plants [12].

Recent studies have corroborated that ROS, RNS, and RSS have similar chemical structures, yet RSS is more versatile and reactive. In addition, ROS/RNS-mediated biosynthesis of RSS under an oxidative environment is its unprecedented source [9]. Due to its similar chemical nature, RSS triggers oxidation reactions by modifying cysteine sulfur and produces an identical effector response as ROS and RNS. Still, the reaction of RSS is more prominent and stable for a much longer duration [13]. Recent discoveries have spectated the growth stimulatory effect of RSS on plant growth and development under stress conditions [6,11,12]. The functional mechanism by which RSS exerts a change stimulatory effect in plants has been unearthed by a few researchers who identified that RSS signaling stimulates post-translational modification of cysteine residues (Cys-SSH) that regulate the expression of stress-responsive genes/proteins [13].

Further, RSS-induced sulfidation, or sulfhydration of proteins, accentuates critical metabolic pathways that stimulate the biosynthesis of secondary metabolites and enzymatic antioxidants, thereby improving the physiology and morphology of plants exposed to climate extremes [12]. Despite their versatility and stability, RSS chemistry, biosynthesis, and plant functions remain anonymous to date. Several excellent review articles have been published on ROS and RNS signaling in plants [9,10]. Discussing them in detail will result in the lengthening of the context. Therefore, readers are requested to consider the articles mentioned above. This review provides an in-depth understanding of RSS biosynthesis in plants and its chemical and biological properties in regulating the functions of proteins/genes and signaling pathways under stress conditions. In addition, this review also examines how RSS interact with other reactive oxidants, such as ROS and RNS, and their combinatorial effect in regulating the growth and metabolism of plants. 

## 2. Chemical Biology of RSS in Plants

In recent years, RSS has been proclaimed to be inexorably interlinked with all life forms from its inception to the present day [14]. Several studies have confirmed that ROS and RSS are chemically similar and are often grouped under the category of chalcogens, i.e., both belong to group 16 of the periodic table [15]. Concomitantly, ROS and RNS mediate the production of RSS under oxidative environments, which is a potential route for RSS biosynthesis [9]. RSS, like ROS, is also categorized as free radicals and non-radicals. Non-radicals are comprised of thiol (RSH), disulfide (RSSR), sulfenic acid (RSOH), and thiosulfate (RSOSR), whereas free radicals are composed of thiyl-radical (RS˙) [14]. Various organisms such as plants, bacteria, fungi yeasts, nematodes, and humans employ different chemical reactions at intracellular and molecular levels, thereby affecting physiological and molecular processes in the concerned organisms [15]. 

During the prebiotic era, hydrogen sulfide (H_2_S) was the primary energy source catalyzing significant steps for incepting life that drove the evolution [16]. In a study, Wachtershauser’s group tried to mimic the conditions before photosynthesis, where organic carbon and nitrogen molecules in the presence of H_2_S catalyze the formation of thiosulfoxide, which is then converted to persulfide via one-electron oxidation reactions [17]. Correspondingly, upon its subsequent exposure to additional thiosulfoxide, persulfide produces polysulfides (H_2_Sn) capable of catalyzing oxidative metabolism in purple and green sulfur bacteria [17]. Thiosulfoxide, persulfide, and polysulfide can be stored and recycled/reduced back to H_2_S to accentuate future oxidation reactions, thus confirming their capability to induce oxidation/reduction reactions for stimulating various signaling pathways [18]. In contrast, ROS and RNS do not possess such ability; they cannot be stored or reused in signaling or related pathways because they function as “one and done”.

However, despite dissimilarity, RSS exhibits higher similarity with ROS than RNS because both are chalcogens with six electrons in their valence shell [19]. Nonetheless, RSS is considered the most versatile, promiscuous, and stable reactive oxidant of its counterparts, i.e., ROS/RNS. Due to higher electronegativity, outer shell electrons in oxygen are near the nucleus, as in sulfur, where electrons are farther from the nucleus [9]. Furthermore, the most stable oxidation state for oxygen is −2 and 0; however, it may also exist in a less stable form of −1, +1, and +2; on the other hand, the most stable oxidation state for sulfur ranges from −2 to +6 [14]. Additionally, sulfur contains more than 30 allotropes compared to oxygen, which has fewer than 10, further attesting to the flexibility of RSS under extreme environments [9]. Strikingly, ROS and RSS exist in various forms with similar chemical and functional properties, i.e., ROS is produced from the one-electron reduction of oxygen (Table 1). In contrast, RSS is produced from one-electron oxidation of H_2_S [10]. ROS, such as hydrogen peroxide (H_2_O_2_), superoxide anion (O_2_˙^−^), hydroxyl radical (OH˙), perhydroxy radical (HO_2_), and singlet oxygen (^1^O_2_), are exclusively involved in signaling under stress conditions [10]. Below, Equation (1) depicts the generation of ROS via a single oxygen reduction process:(1)$$O_{2}\;\overset{+/\_e\_}{↔}\;O_{2}{}^{˙}{}^{-}\;\;\;\overset{+/\_e\_}{↔}\;\;H_{2}O_{2}\;\overset{+/\_e\_}{↔}\;\mathrm{OH}{}^{˙}\;\overset{+/\_e\_}{↔}\;H_{2}O$$

While perhydroxy radical (HO_2_) and singlet oxygen (^1^O_2_) are less intense, signaling molecules often scavenged into peroxide and oxygen are relatively impermeable across membranes. They have a short half-life, are unstable, and are less reactive.

In contrast, `RSS, which are produced by single-electron oxidation of H_2_S, are often composed of thiyl radical (HS˙), hydrogen persulfide (H_2_S_2_), and persulfide radical (S_2_˙^−^) (Table 1). This biosynthetic reaction ends with the formation of elemental sulfur; the steps of which are depicted in Equation (2):(2)$$S_{2}\;\overset{+/\_e\_}{↔}\;S_{2}{}^{˙}{}^{-}\;\;\;\overset{+/\_e\_}{↔}\;\;H_{2}S_{2}\;\overset{+/\_e\_}{↔}\;\mathrm{HS}{}^{˙}\;\overset{+/\_e\_}{↔}\;H_{2}S$$

## 3. RSS Biosynthesis in Plants

In plants, the genus Allium provides an intriguing insight into the biosynthetic pathway of RSS from natural sulfur agents [22]. The first step in RSS production is the biosynthetic transformation of sulfoxides to thiosulfinates, i.e., allin to allicin catalyzed by the alliinase enzyme C-S-lyase type enzyme of 103 kDa with two similar subunits of 448 amino acids [22,23]. The alliinase enzyme contains pyridoxal 5′-phosphate (PLP) at the active site, enhancing the enzyme’s efficiency when the substrate is present at a low level, i.e., allin. In the presence of allin, alliinase catalyzes the formation of highly reactive aldimine, resulting in the generation of sulfenic acid (RSOH) and a PLP amino acylates complex [22]. The complex becomes detached from the RSOH and combines with ammonia and pyruvate to facilitate the formation of thiosulfinate (RSOSR), such as allicin/water [9].

Thiyl radicals, a free radical RSS, are synthesized via single-electron oxidation of the thiol group occurring under oxidative stress conditions, where a proton from the S-atom is removed before or after electron dissemination leading to the formation of disulfide [22]. A notable example of thiyl free radical formation is seen during the glutathionylation of several proteins in plants, where thiyl radical intermediates are provoked by ROS/RNS and other enzymatic antioxidants to generate RSS radicals [23]. Furthermore, researchers have indicated that thiol/disulfide exchange between RSSR and RSH can potentially shift the redox potentials of RSH in a powerful way that can oxidize RS (O) SR from four thiol radicals (R’SH) which are later converted to thiol and R’SSR [24]. 

Thiosulfinates are highly reactive non-radical RSS that can react with any molecule containing the thiol (R’SH) group to form sulfenic acid and disulfide (RSSR) [24]. Additionally, disulfides can be oxidized by themselves under stress conditions or can be corrupted by ROS/RNS to thiosulfinates and later into thiosulfonates (RD (O)2SR), thus validating the involvement of ROS/RNS in RSS biosynthesis [23]. Concomitantly, ROS/RNS-mediated oxidation of the S-atom of sulfenic acids results in sulfinic acid formation. The S-atom of the latter has a higher oxidation state than the former, which can cause further oxidation of sulfinic acid to sulfonyl radicals, which are considered the progenitors of RSS biosynthesis in plants [23]. 

Interestingly, sulfenic acids also have the potential to oxidize thiols to disulfides and, at the same time, can initiate the reduction of cumene and linoleate and, therefore, are also called ultimate reductants [24]. Like superoxide anions, sulfenic acids also possess electrophilic and neutrophilic properties, thus acting as powerful oxidizing and reducing agents leading to the formation of RSS even by reducing sulfur-containing molecules or by oxidizing sulfur atoms from cell proteins, such as cysteine and methionine [13]. A large body of literature has indicated that sulfenic acid-mediated oxidation of protein-thiols could result in the generation of RSOH, RS (O)2H), RS (O)3H, and RSSR [6]. Sulfenic acids are pivotal RSS molecules that can readily react with thiols to form disulfides, exponentially increasing thiol-containing proteins’ catalytic efficiency. In addition, sulfenic acids are included upon hydrolysis of nitrosothiols (nitric oxide donors), modulating critical biological processes and signal transduction. 

## 4. RSS Signaling in Plants

The functional mechanism by which RSS works as a signaling molecule is relatively analogous to that of ROS/RNS signaling; nonetheless, the former is more intricate and perplexing at the biochemical and molecular levels [22]. Several researchers have established that ROS and RSS exploit cysteine residues as the “junction” to mediate redox signaling, sensing, and catalysis of critical metabolic functions [23]. Recent studies have corroborated that RSS function as a signaling molecule after interacting with thiol (-SH) containing proteins (cysteine and methionine), thus regulating persulfidation through post-translational modification (PTMs). Furthermore, RSS can modulate various PTMs such as *S*-glutathionylation, *S*-nitrosation, *S*-cyanylation, and *S*-acylation upon interaction with RNS, glutathione, cyanide, and fatty acids [13]. The profound biochemical implication of RSS in plant cells has delineated its intricate interaction with various antioxidant enzymes, such as glutathione reductase, peroxidase, and sulfite oxidase, thus mediating sulfur metabolism in plant peroxisomes [25]. Correspondingly, this sulfite oxidase, in conjunction with the catalase enzyme, catalyzes the conversion of sulfite to sulfate, thus resulting in the generation of ROS [26]. The existence and inception of RSS have been recently mapped in peroxisomes of root tips and guard cells of Arabidopsis using confocal laser scanning microscopy (CLSM) and proteomic techniques [13,27].

Furthermore, researchers have also mapped the position of corresponding protein genes involved in RSS biosynthesis and observed that the majority of them were present on chromosomes 1, 2, 3, and 5 (Figure 1A,B) along with the protein/genes of ROS and RNS biosynthesis in Arabidopsis [28]. In addition, conserved motif analysis and gene sequence alignments demonstrated that all the reactive species producing enzymes/proteins have a highly conserved arrangement of motifs with varying sequences (Figure 1C and Figure 2A) [29]. The phylogenetic and gene co-expression analysis (Figure 2B,C) of the major reactive species producing enzymes/proteins diverge from one another as they evolve from a common ancestor, and co-expression of the corresponding genes is essential for activating the transcription of stress-responsive genes and transcription factors [28,29]. 

Scientists have only recently uncovered that the peroxisomal enzymes are the primary targets of the persulfidation of thiol-containing proteins leading to the generation of RSS under stress conditions [30]. Later, in vitro analysis in Arabidopsis and *Capsicum annum* unraveled the mechanistic insight into the enzymatic pathway of RSS biosynthesis, where catalase functions both as oxidase and reductase to catalyze the synthesis of H_2_S [30]. Apart from catalase and peroxidase, superoxide dismutase can generate persulfide by stimulating the reaction between O2 and H_2_S in mitochondria and peroxisome, thus validating that apart from ROS/RNS, these organelles also serve as the site of RSS biosynthesis [31]. Still, scientists are trying to delve deeper and further to identify the precise biochemical and metabolic pathway involved in the biosynthesis of RSS in plants, as well as to unravel the mechanism by which they interact with other signaling molecules [13]. Few studies have identified possible enzymatic routes of RSS biosynthesis in the different subcellular compartments of plants, such as cytosol, chloroplast, mitochondria, and peroxisomes. They postulated that the enzymes such as D-cysteine desulfhyfrase, cyano alanine synthase, sulfite reductase, cysteine synthase, and cysteine synthase-like modulate RSS signaling in plants exposed to climate extremes [13,25]. 

Plant peroxisome is the house of various signaling molecules such as glutathione, *S*-nitrosoglutathione, hydrogen peroxide, and sulfite oxidase that are explicitly involved in the RSS biosynthesis and metabolism [32]. Recent discoveries have indicated that RSS signaling profoundly mediated the peroxidation of cysteine sulfur resulting in the formation of sulfenyl via a process known as sulfenylation. Sulfenylation is analogous to persulfidation, a method capable of accentuating RSS-mediated redox sensing and signaling, thus modulating protein folding, histone modification, DNA–protein interaction, and other regulatory functions [33]. One of the mechanisms by which RSS signaling works is that both sulfenylation and persulfidation can reduce hydrogen peroxide to water, which in turn induces the restoration of protein thiols, thus provoking their interaction with other low molecular weight thiols (thioredoxins and reduced glutathione) [34]. The interaction between persulfidated proteins and low molecular weight thiols results in the generation of polysulfide, which is actively involved in the storage and transportation of sulfur molecules capable of initiating RSS signaling [33]. RSS signaling is also incepted upon the NADPH-mediated reduction of persulfidated proteins, which upon interaction with peroxisomal ROS, results in the generation of RSS [35]. Some researchers have also advocated accentuating RSS signaling by incorporating cysteine persulfides into proteins via ROS-mediated PTMs [30]. Furthermore, peroxide signaling is often associated with unresolved conundrums regarding RSS signaling, yet it is hypothesized peroxide signaling acts as a relay between peroxidase and thiol proteins. Correspondingly, the resulting reaction product (oxidants) establishes a complex with the peroxidase proteins, thus categorizing the oxidant signals involved in the modulation of RSS signaling [36].

## 5. Crosstalk between ROS/RNS/RSS in Plant Defense

Plant stress signaling comprehends an intricate network of various signaling molecules that function synergistically or antagonistically to regulate multiple physiological and metabolic processes under specific environmental conditions [37]. Several plant growth regulators, such as salicylic acid, auxin, jasmonic acid, ethylene, abscisic acid, polyamines, and melatonin, are involved in modulating a broad range of plant functions [6]. Apart from these, the gasotransmitters, such as H_2_S and NO, have also been recognized as an indispensable combination for regulating plants’ physiological and stress-related processes (Table 2) with high efficacy and stability compared to their natural counterparts [6]. A molecule can act as a signal only when its level reaches beyond the optimum threshold, driven by complex processes occurring inside the cell. Despite the complex network, all the reactive species, whether ROS, RNS, or RSS, diverge at some metabolic checkpoints where they encounter identical targets responsible for their production, accumulation, and scavenging [18]. When these signaling molecules (they may contain reactive species or plant growth regulators) are exogenously applied, it results in the transient accumulation of compounds that exaggerate the endogenous level of ROS, RNS, or RSS [37]. This transient accumulation of reactive species could have synergistic or antagonistic effects before or after stressful events. 

Recent studies have corroborated that the exogenous application of these signaling molecules is integrally involved in regulating complex networks (Table 2) such as redox status, stomatal movement, root development, and apoptosis in plants exposed to climate extremes [38,39]. However, the effects of these signaling molecules are primarily dependent upon their dosage, subcellular compartment, duration of the application, and their relative flux in the treated plant cell/tissue [40]. A large body of literature exists on the stimulatory role of H_2_S and NO in plants exposed to biotic/abiotic stresses, which has been summarized in recent review articles [6,41,42,43]. Therefore, this crosstalk will provide current updates on the involvement of H_2_S and NO in modulating plant functioning at the molecular level, which is imperative for understanding the role of these gasotransmitters in stress alleviation.

**Table 2 life-13-00204-t002:** Major regulatory effects of H_2_O_2_, NO, and H_2_S on the physiological and metabolic process in plants exposed to climate extremes.

Processes	Functional Mechanisms	Stress Conditions	Pathway Involved	References
Seed germination				
H_2_O_2_	Break seed dormancy and promote seed germination by initiating protein carbonylation	Drought, heat, and salinity	GA signaling pathway	[44,45]
NO	Regulate seed germination and pollen tube growth by activating catabolic enzymes of ABA and GA biosynthesis	Osmotic and heavy metals	ABA and GA signaling pathways	[46]
H_2_S	Promote seed germination of wheat plants by enhancing amylase and esterase activities	Heavy metal and salt	Antioxidant defense pathway	[47,48]
Stomatal movement				
H_2_O_2_	Induce stomatal closure by regulating the activity of NADPH oxidase and control the level of ROS modulates’ stomatal opening by invoking plant peroxisome—specific autophagy	Drought and pathogen attack	ABA and antioxidant signaling pathways	[49,50]
NO	Promote stomatal closure by inducing tyrosine nitration and *S*-nitrosation of ABA receptors	Drought and salinity	ABA and antioxidant signaling pathways	[51,52]
H_2_S	Induce stomatal closure by affecting the activities of ion channels via inducing sulfhydration	Drought and cold	ABA and MAPK signaling	[51,53]
Root organogenesis				
H_2_O_2_	Instigate de novo root organogenesis by acting downstream of auxin and Ca^2+^ signaling	Oxidative stress	Auxin signaling pathway	[54,55]
NO	Modulate root organogenesis by activating the expression of MYB and BHLH transcription factors	Oxidative stress	Auxin and jasmonic acid signaling pathways	[54]
H_2_S	Induce adventitious root formation by acting as upstream of IAA and NO signaling	Heat and heavy metal	Auxin and abscisic acid signaling pathways	[56,57]
Leaf senescence /fruit ripening				
H_2_O_2_	Delay leaf senescence/fruit ripening by regulating the ascorbate-glutathione cycle	Oxidative stress	Antioxidant defense pathway	[58]
NO	Delay induced leaf senescence/fruit ripening by enhancing the expression of stress-responsive genes, ACC synthase, and oxidase enzymes	Salt stress	Antioxidant and ABA signaling pathways	[59,60]
H_2_S	Delay leaf senescence/fruit ripening by regulating the expression of the *Des1* gene	Oxidative stress	Abscisic acid signaling pathways	[56,61]
Post-translational/ epigenetic regulation				
H_2_O_2_	Induce oxidative posttranslational modifications, DNA methylation, and histone modification, thus stimulating plant stress response	Abiotic/biotic stress	Stress defense pathway	[62,63]
NO	Induce tyrosine nitration and *S*-nitrosation and epigenetic regulation of various small and long non-coding RNAs, thereby regulating plant immune system	Abiotic/biotic stress	Stress defense pathway	[63,64]
H_2_S	Induce *S*-Sulfhydration of cysteine residues, thereby activating plant tolerance	Abiotic/biotic stress	Stress defense pathway	[65]

Abbreviation: H_2_O_2_: hydrogen peroxide, NO: nitric oxide, H_2_S: hydrogen sulfide, GA: gibberellic acids, ABA: abscisic acid, ROS: reactive oxygen species, NADPH: nicotinamide adenine dinucleotide phosphate dehydrogenase, MAPK: mitogen-activated protein kinase, BHLH: basic helix loop helix, MYB: myeloblastosis transcription factor, IAA: indole acetic acid, Des1: l-cysteine desulfhydrase, Ash: ascorbate, Gsh: glutathione, ACC: 1-Aminocyclopropane 1-carboxylic acid.

### 5.1. Regulation of Gene Expression

The functional analogy of ROS, RNS, and RSS has been extensively studied by applying them exogenously to analyze their potential in modulating growth and defense-related processes in plants exposed to environmental cues [18]. Recent studies have shown that both NO and H_2_S are potentially involved in regulating seed dormancy and germination via modulating ABA signaling [51]. Molecular investigation revealed that NO exerts this effect by activating PYR/PYL/RCAR (pyrabactin resistance 1/PYR, such as the regulatory component of ABA receptor), SnRK2 (SNF1-related protein kinase), and ABI5 (abscisic acid insensitive 5) protein, whereas H_2_S signaling stimulated seed germination in Arabidopsis by modulating the expression of L-DES (L-cysteine desulfhydrases) protein [66,67]] Recent investigations in Arabidopsis indicated that H_2_S signaling results in the accumulation of constitutive photomorphogenesis 1 (COP1) in the nucleus that stimulates the degradation of elong hypocotyl 5 (HY5). The increased degradation of HY5 resulted in decreased expression of the ABI5 gene, thereby enhancing seed germination under heat stress [68]. In another study, the combinatorial effect of NO and H_2_S signaling regulated the adventitious rooting in tomato plants by activating the expression of auxin-related genes (ARF4 and ARF16) and cell cycle-related genes (CYCD3, CYCA3, and CDKA1) [69]. Exogenous application of NaHS leads to the activation of transcription and activity of L-DES1 that stimulates the H_2_S biosynthesis, which in turn activates the germination of Arabidopsis seeds [70]. In mutant (lcd/des1 defective) plants, exogenous application of NaHS does not induce the germination of the Arabidopsis seeds. In wild plants (lcd/des1 induced), NaHS-mediated increase expression of L-DES1 protein activated the H_2_S biosynthesis have successfully maintained the GSH/GSSG ratio within the cells that may have triggered alternative oxidase (AOX) mediated cyanide resistant respiration to regulate seed germination [70]. Similar observations have also been made where the exogenous application of NO and H_2_S positively modulated the seed germination and lateral root formation in cucumber and maize plants by triggering the expression of the heme oxygenase-1 (HO-1) gene and by reorganizing the arrangement of F-actin bundles [71,72,73]. 

Another complex process that is sophistically regulated via signaling molecules (ABA, ethylene, and K^+^/Ca^2+^) is the opening/closing of stomata, and recent investigations have confirmed the role of NO and H_2_S signaling in controlling guard cell function [67,74]. Previous investigations have concluded that NO signaling modulates stomatal movement by regulating the expression of SNF-1 (sucrose nonfermenting 1) and SnRK2.6 genes [75]. In contrast, H_2_S signaling negatively regulated the opening of stomata via inducing NO-mediated activation of 8-mercapto-cGMP, which restricts the inward flow of K+ in guard cells [76,77]. Recent investigations have reported H_2_S signaling downstream of mitogen-activated protein kinase 4 (MAPK4), thereby inducing ABA-mediated stomatal closure in response to drought stress [78]. A study concluded that the exogenous application of NaHS significantly altered the drought-tolerant capacity of mapk4 and slac1 mutant plants compared to wild-type Arabidopsis plants. H_2_S signaling could not effectively regulate the lcd/des1 gene expression, which has affected the ABA-mediated stomatal movement in mutant plants [78]. Similarly, the exogenous application of H_2_O_2_ induced H_2_S signaling by activating the expression of the L-DES1 gene significantly regulated the salt stress tolerance in *Vicia faba* plants by altering the movement of guard cells [79]. Recently, researchers have confirmed that the NO can modulate ABA-induced stomatal closure by activating the GSNO reductase gene, which in turn causes differential regulation of nitrate reductase genes (NIA1 and NIA2) and CLE 9 peptide (Clavato3/Embryo Surrounding region) [52]. The differential accumulation of NIA genes alters the ABA-induced activity of K^+^ inward and activates the outward flow of K+ in wild-type Arabidopsis plants. In contrast, the down-regulation of NIA genes abolishes the inward movement of K^+^ [52].

Several studies have demarcated NO and H_2_S signaling in leaf senescence and fruit ripening. Previous works have reported a substantial role of exogenous NO in reducing ethylene production and ABA-induced senescence in rice and strawberry plants [80,81]. NO regulated ethylene production and senescence by modulating the expression of NOS-like enzymes, leading to differential accumulation of endogenous NO [82]. Later, researchers indicated that exogenous application NO evidently interacts and activates SA and JA signaling, which induces the expression of NOS-like enzymes, thereby delaying leaf senescence upon upregulation of the antioxidant defense system [83]. Recently, H_2_S signaling in regulating leaf senescence was analyzed by inducing mutation in an H_2_S-producing enzyme encoding L-cysteine desulfhydrase 1 (L-DES1). They concluded that mutant (Des1) Arabidopsis plants showed enhanced susceptibility to drought stress and accelerated leaf senescence [84].

In contrast, over-expressed Arabidopsis plants (OE-DES1) exhibited more tolerance to drought and delayed leaf senescence [84]. The effect of LCD1 on the regulation of leaf senescence was explored in wild-type, mutant, and overexpressed tomato plants. The mutant LCD1 plants showed early development of leaf senescence, whereas overexpressed LCD1 plants exhibited delayed leaf senescence compared to wild-type tomato plants. Further investigations revealed that the delayed leaf senescence in overexpressed tomato plants was due to the downregulation of various chlorophyll degradation genes, such as NYC1, PAO, PPH, and SGR1, and senescence-associated genes (SAGs), along with active scavenging of ROS [85]. 

Fruit ripening is governed by a complex physiological process fine-tuned via various signaling molecules, plant growth hormones, and environmental stimuli [86]. In recent years, extensive studies have been conducted to unravel the mechanism of regulation of fruit ripening mediated via reactive species (ROS/RNS/RSS). Exogenous application of NO and H_2_S differentially regulated the endogenous level of NO and H_2_S, accentuating fruit ripening in *Capsicum annum* fruits. The differential regulation of endogenous NO and H_2_S levels altered the activity of the NADP isocitrate dehydrogenase enzyme, thus promoting fruit ripening in pepper plants [87]. A year later, they conducted a different study to reveal that an increase in the endogenous NO and H_2_S levels differentially regulated the fruit ripening process in *Capsicum annum* fruits by downregulating the expression of the NADP malic enzyme and the 6-phsphogluconate dehydrogenase enzyme [88].

Furthermore, Zhu et al. [89] reported the synergistic effect of NO and H_2_S on the inhibition of peach fruit ripening during storage. They concluded that the combined application of NO and H_2_S caused a significant reduction of ethylene production, i.e., it decreased the expression of ACC synthase and ACC oxidase enzyme and cell wall metabolism to delay fruit ripening in peach plants. Interestingly, the exogenous application of strigolactones regulated the endogenous level of NO and H_2_S by upregulating NOS-like enzymes and L-cysteine desulfurase, which concomitantly delayed the ripening of strawberries [90]. They further concluded that the delayed development of strawberries was due to differential regulation of the antioxidant defense system mediated via increased NO and H_2_S during storage.

### 5.2. Regulation of Post-Translational Modification

Recent research has confirmed that ROS/RNS/RSS regulates the expression of various stress-responsive genes and modulates specific proteins’ activity via post-translational modifications (PTMs) [6]. PTMs such as Persulfidation, tyrosine nitration, S-nitrosylation, and carbonylation are activated upon exogenous application of RSS/RNS/ROS, thereby influencing plant growth and developmental processes under stress conditions [37]. Cysteine thiols are a highly nucleophilic group that regulate a plethora of protein regulatory functions, such as their catalytic efficiency, stability, and interaction with ligands upon reacting with ROS/RNS/RSS [91]. Persulfidation, previously known as *S*-sulfhydration, is a reversible oxidative PTM process where the thiol group (-SH) is converted into persulfide form (-SSH) in a complex reaction, in which protein thiol reacts with an anionic and protonated form of H_2_S or reactive species-mediated interaction of protein thiols with inorganic polysulfides [91]. The advent of ultra-throughput proteomic approaches and advancement in mass spectrophotometry has led to the identification of several proteins undergoing persulfidation [91]. Exogenous application of H_2_S (NaHS) coupled with proteomic analysis identified 1000 differentially regulated proteins in Spinacia oleracea plants, of which few were persulfidated [92]. Similarly, researchers employed the biotin switch method with liquid chromatography–tandem mass spectrometry (LC-MS). They identified 106 to 2015 (using the modified switch method) persulfidated proteins in Arabidopsis leaves [12]. In addition, exogenous H_2_S persulfidated the mitogen-activated protein kinase 4 (MAPK4), which causes up to a 10-fold expression of MAPK4, leading to the enhanced tolerance of Arabidopsis plant’s chilling stress [93]. Concomitantly, the exogenous application of NaHS reverses the adverse effect of heavy metals inducing oxidative damage in tomato plants by inducing the persulfidation of enzymatic antioxidants such as CAT, APX, GR, and SOD [94]. H_2_S-mediated persulfidation significantly regulated the expression of the L-DES1 gene, which modulated the ABA signaling in guard cells, leading to the opening and closing of stomata [68]. They further concluded that H_2_S mediated persulfidation of Open Stomata 1/SNF1-related protein kinase 2.6 (SnRK2.6) increases its ability to interact with another transcription factor, such as DREB and ABA-responsive factor 2 (ABF2), to promote stomatal closure [68]. Interestingly, H_2_S-mediated persulfidation of ABI4, RBOHD, L-DES1, ATG4, and ATG18a have improved plant cellular and physiological processes under stressful conditions by preventing reversible protein oxidation and DNA damage [95,96,97]. 

*S*-nitrosation is a PTM process essentially regulating discrete signaling functions in plants ranging from seed germination to fruit maturation and is also mechanistically involved in regulating abiotic/biotic stress tolerance [51,98,99]. Much of the NO-derived PTMs in plants are accentuated by tyrosine nitration followed by *S*-nitrosation, which involves adding nitric oxide radical to protein thiols to generate *S*-nitrosoglutathione (GSNO) [13,64]. Extensive research on NO-mediated PTMs has been conducted in Arabidopsis plants, where the researchers have identified several *S*-nitrosated proteins after treating them with a NO donor (GSNO) [100]. Similar reports have also been published [101,102,103] where they used a combination of BSM and affinity tags to identify targets of *S*-nitrosation proteins in Arabidopsis plants exposed to biotic and abiotic stresses. *S*-nitrosation and tyrosine nitration was involved in heavy metal-stressed mutant and wild-type Arabidopsis plants [104]. In their study, the researcher identified that *S*-nitosothiol signaling was responsible for enhancing NADPH-dependent thioredoxin reductase activity, which is involved in the regulation of hydrogen peroxide, and further application of LC-MS/MS analysis indicated that heavy metal-induced *S*-nitrosation of APX resulted in enhanced tolerance to stressed plants [104]. Ripening of *Capsicum annum* fruits is also differentially regulated by NO-mediated PTMs [105]. They exploited BSM, nano-liquid chromatography, and mass spectrometry to identify that Cys377 of peroxisomal catalase potentially undergoes *S*-nitrosation, thereby modulating ROS/RNS levels in fruits.

Further, Silva et al. [106] identified potential targets and substrates for *S*-nitrosation in plants. Their study determined that cytosolic and plastid glutamine synthetase is differentially regulated by *S*-nitrosation, where Cys369 is a possible site. Later, [107] postulated that the exogenous application of NO-mediated nitration of tyrosine residues and *S*-nitrosation of cysteine residues differentially affected vital amino acid residues involved in the binding of FAD and molybdenum cofactors in Arabidopsis plants. Conversely, *S*-nitrosation of protein thiols has modulated tomato plants’ stress defense response against *Phytophthora infestans* [108]. Their study identified decreased activity of the *S*-nitrosoglutathione reductase enzyme in both tolerant and susceptible genotypes.

In contrast, *S*-nitrosation protein was increased in the tolerant genotypes as the infection progressed. Correspondingly, proteomic approaches identified tolerant plants exhibiting increased protein-*S*-nitrosation and showed enhanced accumulation of osmolytes, metabolites, and enzymatic and non-enzymatic antioxidants [108]. Recently, time-course gene expression analysis and non-denaturing PAGE have confirmed that PTMs such as tyrosine nitration, *S*-nitrosation, and glutathionylation heterologously regulated the expression of APX enzyme in the fruits of *Capsicum annum* plants [109]. 

Carbonylation is an irreversible PTM mediated by various ROS such as hydroxyl radical (HO˙), superoxide anion (O_2_˙^−^), and hydrogen peroxide (H_2_O_2_) that potentially add aldehyde/ketone group into side chains of specific amino acids [110]. Carbonylation is a complex chemical process driven in conjunction with lipid/sugar derivatives and is basically of two types, viz. primary/secondary carbonylations [111]. Primary protein carbonylation is often modulated via metal-associated oxidation of Lys, Pro, and Thr residues or through α-amidation of glutamyl residue to form aldehyde and ketone derivatives [110]. Conversely, Secondary protein carbonylation is driven by adding reactive carbonyl species (RCS) to Cys, His, and Lys residues of specific proteins [112]. A large body of literature has confirmed that the protein carbonylation is exaggerated upon an increase in cellular ROS levels that differentially regulate plant cell growth and physiology under stress conditions [111,112]. The effect of protein carbonylation has been profoundly studied in Arabidopsis plants, where researchers have reported that the protein carbonylation first increases and then decreases with the age of the plants, thereby drastically affecting the reproductive and senescence phase [113]. Researchers used liquid chromatography–tandem mass spectrometry to identify carbonylation sits in *Phaseolus vulgaris* plants [114]. They further concluded that approx. 238 proteins were carbonylated, of which maximum effect was observed in two essential nodule proteins, i.e., glutamine synthetase and glutamate synthase, that conversely regulated cell metabolism and nodule senescence in legume plants [114].

Furthermore, a proteomics approach was employed to identify carbonylated proteins in premature, mature, and desiccated seeds of *Medicago truncatula* [115]. The PM34 protein, a protein responsible for seed growth and also known to possess cellulase activity, was carbonylated more readily than other proteins. Effect of As accumulation on protein carbonylation was analyzed in *Oryza sativa* root and shoots [116]. Researchers concluded that increased As accumulation drastically enhanced lipid peroxidation and protein carbonylation, affecting rice plants’ yield. Likewise, ROS-mediated protein carbonylation of enzymatic and non-enzymatic antioxidants invariably decreases seed vigor and other physiological activity of the seed of wheat plants [117]. Similarly, [118] identified 180 carbonylated proteins in ABA-treated *Arabidopsis thaliana* leaves using affinity chromatography that differentially regulated stomatal closure upon ABA supplementation. 

*S*-acylation, previously known as *S*-palmitoylation, is a reversible PTM that instigates the palmitate group’s addition to the specific Cys residues of soluble/membrane proteins [119]. *S*-acyl transferases are the main catalyzing this reaction, having conserved the Asp-His-His-Cys motif and formed a thioester bond in the target proteins [120]. Numerous studies have identified and quantified *S*-acylated proteins in Arabidopsis and poplar plants using a proteomics approach coupled with BSM labeling [119,121]. Researchers have corroborated that the proteins which are most commonly *S*-acylated in plants growing under climate extremes are calcium-dependent protein kinases (CDPK), mitogen-activated protein kinases (MAPK), ATPases, calcineurin-B-like proteins, and several integral proteins [122,123]. *S*-acylation of several receptors, such as kinases proteins adjacent to the transmembrane domain, differentially regulated the flagellin perceiving receptor kinases 2 (FLS2) and associated signaling function [124]. Interestingly, another study concluded that *S*-acylation of plant immune receptor proteins such as P2K1, FLS2, and CERK1 activated the defense response of Arabidopsis plants more prominently compared to their corresponding non-mutant plants [125]. Recently, *S*-acylation of Alpha/Beta Hydrolase domain-containing protein 17-like acyl protein thioesterases invariably affected cell death/apoptosis in Arabidopsis and *Nicotiana tabaccum* plants [126]. They further concluded that protein-*S*-acyltransferases and de-*S*-acylation enzymes are progressively involved in regulating subcellular localization, stability, and activity of enzyme proteins. Likewise, the *S*-acylation of FLS2 controlled and stabilized the ligand-induced receptor kinase complex, thereby triggering plant innate immune response against pathogen attack [127]. *S*-acylation-mediated regulation of plant immune receptors (DORN1, LecRK-1.9, and ATP receptors) inhibited bacterial invasion in Arabidopsis plants via P2K1-mediated autophosphorylation and protein degradation [128].

### 5.3. Regulation of Epigenetic Modification

Epigenetic modifications, such as DNA methylation and histone modifications, are potentially involved in the modulation of growth, development, and defense process of plants exposed to climate extremes [129]. The epigenetic modifications exert these stimulatory/inhibitory functions by regulating gene expression or performing chromatin remodeling dynamics strongly influenced by ROS/RNS/RSS signaling [129,130]. Accumulating shreds of evidence have indicated that reactive species can potentially affect the post-replication modification of DNA molecules by methylating cytosine at the five positions or adenine at the N4 or N6 positions [131]. Several excellent review papers have comprehensively described plant methylation and histone modifications [129,130,131]. Therefore, this section is mainly confined to ROS/RNS/RSS mediated regulation of epigenetic modifications influencing plant growth and development under stress conditions. These reactive species accentuate epigenetic mechanisms of gene regulation by modulating the functioning of various proteins/enzymes such as histone deacetylases, DNA methylases, Demeter1, a repressor of silencing 1 and Demeter-like 2 and 3 [132,133]. 

Researchers have corroborated that these reactive species directly interact with the Fe-S cluster of Demeter proteins, resulting in the generation of oxidative stress in plants, thus revealing the linkage between redox signaling and epigenetic modification [134]. Some of the pioneering studies in redox regulation of epigenetic modification in transgenic rice plants indicated that plants over-expressing OsSRT1 (silent information regulator 2 (SIR2)-related HDAC gene) exhibited less DNA methylation and oxidative damage compared to wild-type plants [135]. A decade later, Zhang et al. [136] confirmed the above notion and showed that OsSRT1 enhanced the tolerance to oxidative of transgenic plants by differentially regulating the expression of glyceraldehyde-3-phosphate dehydrogenase that had prevented the excessive generation of reactive species, thereby inhibiting DNA damage. Since then, a large body of literature has confirmed that differential regulation of ROS/RNS/RSS signaling can modulate epigenetic modification, such as acylation and nitrosation in Arabidopsis and rice plants via reactive species-mediated development of oxidative stress and decreased expression of HDA19 and HDA9 [33]. 

Likewise, the exogenous application of salicylic acid and nitric oxide accentuated ROS scavenging, thereby inducing DNA de-methylation and enhancing high-temperature tolerance in *Lablab purpureus* L. plants [41,42]. Reactive species-mediated PTMs, such as acylation and nitrosation, have also been implicated in regulating secondary metabolite synthesis and the formation of crown roots in rice plants [33]. In another study, the exogenous application of nitric oxide efficiently modulated the ROS/RNS signaling, thereby improving the tolerance of lettuce plants exposed to metal stress by decreasing DNA methylation and activating the expression of stress-responsive genes [137]. Furthermore, the knockdown of argonaute 2 (OsAGO2) protein in rice anthers exaggerated ROS biosynthesis leading to increased methylation levels, abnormal growth, and apoptosis [138]; whereas an overexpression of hexokinase 1 (OsHXK1) differentially regulated the ROS level and tapetal apoptosis, leading to a significant decrease in the methylation level of the concerned promoter. 

An increasing body of literature has demonstrated DNA methylation’s involvement in fruit ripening, which was analyzed recently by [139]. They concluded that the treatment of grape plants with azacitidine significantly lowered the methylation rate in grapefruit compared to non-treated plants and enhanced the expression of the superoxide dismutase enzyme, thus establishing a link between ROS metabolism and epigenetic regulation. Recently, non-thermal plasma (NTP) treatment of seedlings of *Glycine max* displayed alteration in methylation patterns [140]. Researchers employed two doses of NTP that differentially regulated the ROS/RNS mediated methylation pattern that accentuated seed vigor and germination, resulting in improved seedling growth. In addition, genome-wide methylome status under the influence of H_2_O_2_ production was analyzed in transgenic tobacco plants [141]. They identified that overexpressing the CchGLP gene of transgenic plants stimulated endogenous ROS/RNS signaling, thereby enhancing its biotic and abiotic stress tolerance. Several recent findings have argued and confirmed the multifaceted role of ROS/RNS/RSS in epigenetically regulating gene expression and stress tolerance in plants. Still, this regulation’s molecular mechanism is far-fetched. Therefore, future efforts must be diverted toward unraveling the functional mechanism by which ROS/RNS/RSS strengthens the plant’s innate immunity against biotic/abiotic stresses.

## 6. Potential Biotechnological Application in Agriculture

In the recent decade, the plant science community has diverted its attention toward un-earthing the functional mechanisms by which ROS/RNS/RSS play a substantial role in regulating plant growth, development, and stress amelioration (Figure 3) [37]. Researchers have confirmed that these ROS/RNS/RSS regulate important cellular, physiological and biological processes in plants after interacting with other phytohormones [4,7]. A large body of literature has contemplated the role of NO during seed germination, fruit development, maintaining ion/osmotic homeostasis, and coping with biotic/abiotic stresses [6,11,13]. Furthermore, NO has also been known to differentially regulate the virulence of bacterial and fungal pathogens by acting as an antimicrobial or antifungal agent [142]. In addition, studies have also delineated that the perception of NO gas by microorganisms, such as bacteria and fungi, can induce transcriptional reprogramming of genes involved in metabolic and virulence systems, thereby regulating their adaptation [143]. Correspondingly, the genes involved in NO perception and signaling have been identified in both gram-positive and gram-negative bacteria [142]. The potential biotechnological application of NO in agricultural or horticultural crops depends on the donor molecules, application, tissue type, environment, and possible interaction with other secondary metabolites [142]. The most frequently used NO donor is sodium nitroprusside, and the application of foliar, seed soaking, or irrigation has tremendously improved plant growth and development under stress conditions [41]. Increasing evidence has contemplated that exogenous application of NO (Figure 3) can explicitly break seed dormancy and improve seed vigor, an essential prerequisite for attaining better growth and increasing crop yield [144,145]. Several studies have found that applying NO donors can effectively modulate seed dormancy even at low concentrations compared to the mechanical and chemical methods of breaking seed dormancy, such as H_2_SO_4_ and NaOCl [146,147]. 

Recent investigations have shed light on the functional mechanisms by which NO mediates regulation of seed dormancy/germination by activating PTMs, such as tyrosine nitration of abscisic acid receptors, such as *S*-nitrosylation SnRK2s and ABI5, followed by degradation of ethylene-responsive factors [66]. In another study, Nagel et al. [146] identified novel loci in barley plants capable of regulating NO signaling, thereby regulating seed dormancy and preharvest sprouting. They concluded that the HvGA20ox1 gene is upregulated by NO signaling, alleviating dormancy and initiating flowering in barley plants [146]. Similarly, the significance of H_2_S in helping seed dormancy has been reported by various researchers where exogenous H_2_S upregulated the expression of COP1/HY5 stimulated seed germination in heat-stressed Arabidopsis plants [68]. The implication of exogenous application explicitly improves seed germination and seedling growth in heat-stressed *Zea mays* by regulating the expression of antioxidants and osmolyte biosynthesis [148].

Interestingly, the exogenous application of H_2_S resulted (Figure 3) in the persulfidation of ethylene biosynthetic enzymes, thereby decreasing the sensitivity of ethylene receptors and leading to enhanced seed germination due to the expression of antioxidative enzymes under stress conditions [149]. Recently, the antagonistic effect of the exogenous application of H_2_S has also been observed by [18]. The authors confirmed that H_2_S-mediated persulfidation has also improved seed dormancy/germination by downregulating the expression of DES1 and ABI4 genes.

NO is also known to regulate fruit ripening and post-harvest shelf life by antagonizing ethylene signaling and contributing to shelf life extension [150]. Exogenous application of NO in post-harvest stages of *Magnifera indica* showed decreased membrane damage, ethylene production, and polygalacturonase activities, prolonging the fruits’ shelf life without losing quality [151]. Likewise, the exogenous application of NO positively regulated the physiological weight loss, antioxidants, lipooxygenase, and pectin methylesterase activities, improving the quality/shelf life of *Prunus persica* fruits [152]. In addition, exogenous application of H_2_O_2_ and NO to guava and tomato fruits exhibited delayed ripening and minimum loss in TSS and ascorbic content, thus improving the storage life of these fruits under storage conditions [153]. Similarly, the exogenous application of H_2_S has been demarcated to improve the shelf life and quality of tomato post-harvest fruits by inhibiting the protease activity and decreasing the ERF and beta-amylase gene expression with higher nutritional contents [154]. In another study, exogenous application of H_2_S differentially regulated the expression of cell wall modifying related genes and critical genes involved in pectinase enzyme, thereby preventing rapid fruit softening in the fruits of *Fragaria chiloensis* [155]. Recently, exogenous application of H_2_S on the activity of various biochemical markers linked to fruit ripening in the fruits of banana and pointed gourd in the post-harvest stage was analyzed to detect their quality and shelf life. They unraveled that exogenous application of H_2_S significantly lowered membrane damage, lignification, and chlorophyll loss and correspondingly maintained higher PAL and PPO activity, thereby increasing the shelf life of the fruits without compromising their quality and extending their marketing duration [156,157]. 

NO and H_2_S have been known to modulate legume rhizobia symbiosis, a critical process in agriculture, as it accounts for the availability of biological nitrogen in the soil [158]. NO is believed to regulate two crucial aspects of nitrogen fixation in legume plants (i) by controlling the rhizobial infection and (ii) by activating nodule organogenesis, both of which are significantly regulated upon the expression of nodulation factors (nod genes) [158]. Increasing evidence has confirmed the regulatory role of NO as a signaling molecule for promoting legume–bacteria interaction and nodule development [159,160,161,162]. Exogenous application of NO to *Phaseolus vulgaris* and *Glycine max* stimulated the nodule formation, as indicated by the over-accumulation of nod factors [163]. In addition, studies have suggested that NO and ROS positively regulate legume–rhizobia interaction and nodule formation via ROS/RNS-mediated differential expression of ROS/RNS-producing enzymes [163].

Furthermore, the exogenous application of NO and H_2_S compellingly enhanced the heavy metal stress tolerance of legume–rhizobia symbiosis by increasing the soil enzyme activity and bacterial diversity, thereby improving their growth and productivity [158]. Likewise, the exogenous application of NO regulated the expression of phytoglobin 1.1 in *Medicago truncatula*, accentuating the symbiosis by controlling nodule development and nitrogen fixation [158]. ROS/RNS/RSS amplifier effect in stimulating legume–rhizobia symbiosis is mainly due to the various PTMs catalyzed by these reactive species that activate essential enzymes, such as thiol peroxidase and phytoglobin genes, that efficiently form nodules in legume crops [6]. Future application of these reactive species in a precise and controlled way can potentially regulate crop growth and productivity by regulating many distinct physiological and biological processes. 

## 7. Conclusions

Earlier, ROS/RNS/RSS has been described as toxic molecules affecting various metabolic and regulatory functions in plants exposed to several biotic and abiotic stress conditions. With the advent of technological breakthroughs, different redox biologists have implicated their role in regulating critical cellular and metabolic processes in plants exposed to climate extremes. When these redox regulators are maintained below a threshold level, they control many signal transduction pathways by mediating PTMs and epigenetic modification in plants. PTMs, such as carbonylation, *S*-nitrosation, tyrosine nitration, and persulfidation, instigate dynamic signaling of different components of enzymatic and non-enzymatic defense systems at gene and protein levels. The precise regulation induced by these redox regulators at the biochemical and molecular level strengthens plant innate immune response, thereby maintaining the growth and productivity of plants under adverse growing conditions.

Moreover, researchers have also reported that these redox molecules essentially regulate DNA methylation and histone modification in plants, thus significantly impacting the transcription and activities of various genes and enzymes. Nevertheless, the precise mechanisms by which ROS/RNS/RSS regulate these epigenetic marks and other biochemical and molecular processes under stress conditions need further investigation. Furthermore, in-depth explorations of how these redox regulators interact with other signaling molecules and plant growth regulators must be unraveled. The unraveling of molecular mechanisms underlying the regulatory functions of ROS/RNS/RSS will open a new realm for formulating new biotechnological strategies for their possible application in agriculture for fostering new/improved cultivars in the era of climate change.

## Figures and Tables

**Figure 1 life-13-00204-f001:**
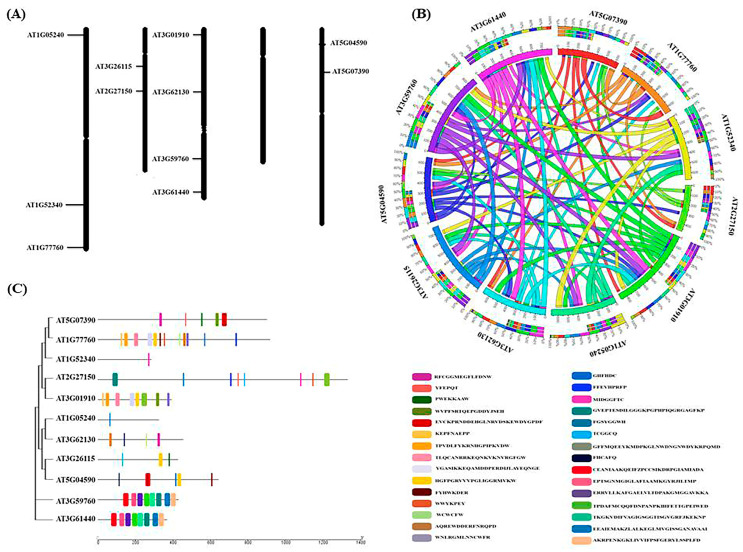
Chromosomal localization, synteny analysis, and motif elucidation of important ROS-, RNS-, and RSS-producing genes in model plant *Arabidopsis thaliana*. (**A**) Chromosomal localization, (**B**) synteny analysis, and (**C**) conserved motifs are represented in different color boxes. AT5G07390/RBOHA: respiratory burst oxidase homolog protein A, AT1G77760/NIA1: nitrate reductase 1, AT1G52340/ABA2: ABA xanthoxin dehydrogenase, AT2G27150/AAO: abscisic-aldehyde oxidase, AT3G01910/SOX: sulfite oxidase, AT1G05240/POX: peroxidase 1, AT3G62130/LDES: L-cysteine desulfhydrase, AT3G26115/DDES: D-cysteine desulfhydrase 2, AT5G04590/SIR: assimilatory sulfite reductase, AT3G59760/CYSC1: cysteine synthase, AT3G61440/QASC: nifunctional L-3-cyanoalanine synthase/cysteine synthase C1.

**Figure 2 life-13-00204-f002:**
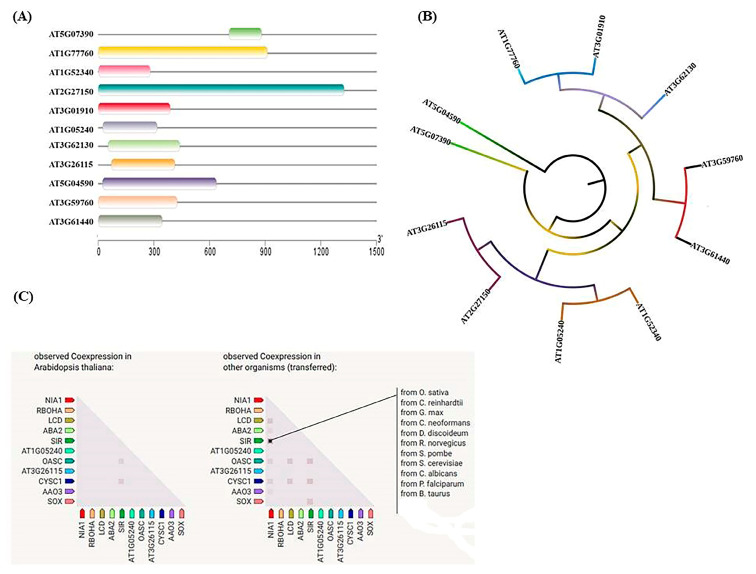
Basic sequence analysis, phylogenetic relationship, and co-expression prediction of important ROS-, RNS-, and RSS-producing genes in the model plant *Arabidopsis thaliana*. (**A**) Gene sequence, (**B**) phylogenetic analysis, and (**C**) co-expression of putative genes using TBtools and iTOL software. AT5G07390/RBOHA: respiratory burst oxidase homolog protein A, AT1G77760/NIA1: nitrate reductase 1, AT1G52340/ABA2: ABA xanthoxin dehydrogenase, AT2G27150/AAO: abscisic-aldehyde oxidase, AT3G01910/SOX: sulfite oxidase, AT1G05240/POX: peroxidase 1, AT3G62130/LDES: L-cysteine desulfhydrase, AT3G26115/DDES: D-cysteine desulfhydrase 2, AT5G04590/SIR: assimilatory sulfite reductase, AT3G59760/CYSC1: cysteine synthase, AT3G61440/QASC: bifunctional L-3-cyanoalanine synthase /cysteine synthase C1.

**Figure 3 life-13-00204-f003:**
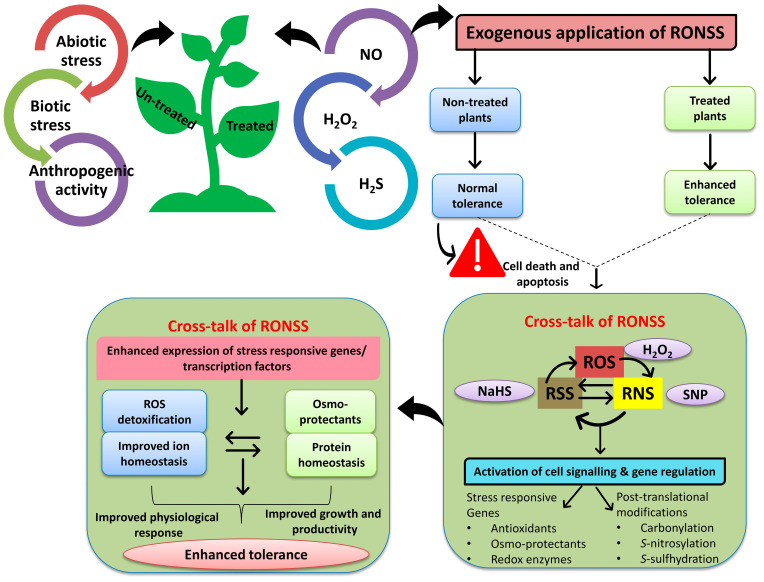
Crosstalk of ROS, RNS, and RSS in cell signaling, gene regulation and their role in ameliorating abiotic/biotic stress tolerance in plants. Exogenous application of SNP (NO donor), NaHS (H_2_S donor), and H_2_O_2_ results in the activation of a plethora of signaling pathways, thus regulating cell signaling and activation of several downstream stress-responsive genes/transcription factors leading to improved stress tolerance.

**Table 1 life-13-00204-t001:** The enzymes involved in the generation of ROS, RNS, and RSS in plants.

	Enzymes	Reaction Catalysed	Cellular Location	Uniprot/Gene	References
**ROS**	Cytochrome C oxidases and alternative oxidases	$O_{2}\;\overset{+/\_e\_}{↔}\;O_{2}{}^{˙}{}^{-}$ (Mehler’s reaction)	Thylakoid membrane	A0A1P8AZ61 /AT2G43780, Q39219/AT3G22370	[20]

	Cytochrome C oxidases and alternative oxidases	O2˙−+Fe3+→O2˙1+Fe2+	Thylakoid membrane/mitochondria	A0A1P8AZ61 /AT2G43780, Q39219/AT3G22370	[20]
	Cytochrome C oxidases and superoxide dismutase	$2O_{2}{}^{˙}{}^{-}+2H^{+}\to O_{2}+\;\;H_{2}O_{2}{\mathrm{Fe}}^{3+}$ (Haber- Weiss reaction)	Thylakoid membrane/peroxisomes	A0A1P8AZ61 /AT2G43780, P24704/AT1G08830	[20]
	Cytochrome C oxidases and alternative oxidases	${\mathrm{Fe}}^{2+}+H_{2}O_{2}\to {\;\mathrm{Fe}}^{3+}+{\mathrm{OH}}^{-}+OH^{{}^{˙}}$ (Fenton reaction)	Thylakoid membrane	A0A1P8AZ61 /AT2G43780, Q39219/AT3G22370	[20]
**RNS**	Nitrate reductase	${\mathrm{NO}}_{2}{}^{˙}+\mathrm{NADPH}\to NO{}^{˙}$	Peroxisomes	P11035 /AT1G37130,	[8,21]

	NOS-like activity, aldehyde oxidase, sulfite oxidase, and xanthine dehydrogenase	$L-\mathrm{Arg}+\mathrm{NOS}\;\mathrm{cofactors}\;\left(\mathrm{FAD},\;\mathrm{MOCO}\right)\to NO{}^{˙}$	Chloroplast, mitochondria	Q7G191/AT1G04580, Q8GUQ8 /AT4G34890	[8,21]
	Peroxidase	$\mathrm{Hydroxyurea}+H_{2}O_{2}\to NO{}^{˙}$	Chloroplast, mitochondria	Q9SMU8/AT3G49120	[8,21]
	Amidoxime reducing components	${\mathrm{NO}}_{2}{}^{˙}+\mathrm{Cyt}\;C\;\left(\mathrm{red}\right)\to NO{}^{˙}$	Chloroplast, mitochondria	LOC9299625	[8,21]
**RSS**	L-cysteine desulfhydrase	$L-\mathrm{Cys}+{\;H}_{2}O\to \;H_{2}S+{\mathrm{NH}}_{3}+\mathrm{pyruvate}\;$	Cytosol	Q9MIR1/At3g62130	[11]

	D-cysteine desulfhydrase	$D-\mathrm{Cys}+{\;H}_{2}O\to \;H_{2}S+{\mathrm{NH}}_{3}+\mathrm{pyruvate}$	Mitochondria	A1L4V7/At3g26115	[11]
	Sulfite reductase	${\mathrm{SO}}_{3}^{-2}+\;e-\to \;H_{2}S+3H_{2}O$	Chloroplast	Q9LZ66/At5g04590	[11]
	Cyanoalanine synthase	$L-\mathrm{Cys}+\mathrm{hydrogen}\;\mathrm{cyanide}\to \;H_{2}S$	Mitochondria	Q9S757/At3g61440	[11]
	Cysteine synthase 1	$L-\mathrm{Cys}+\mathrm{acetate}\to \;H_{2}S$	cytosol	P47998/At4g14880	[11]

Abbreviations: O_2_: molecular oxygen, O_2˙_^−^: superoxide anion, Fe^3+^: ferric ion, ^1^O_2_: singlet oxygen, Fe^2+^: ferrous ion, H_2_O_2_: hydrogen peroxide, OH^−^: hydroxyl anion, OH˙˙: hydroxyl radical, NO_2_˙: nitric dioxide, NADPH: nicotinamide adenine dinucleotide phosphate dehydrogenase, NO˙: nitric oxide radical, L-Arg: L-arginine, FAD: flavin adenine dinucleotide, MOCO: molybdenum cofactor, Cyt C: cytochrome C, L-Cys: L-cysteine, H_2_S: hydrogen sulfide, NH_3_: ammonia, ${\mathrm{SO}}_{3}^{-2}$: sulfite ion.

## Data Availability

Not applicable.

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
