# Peer review of "Free Radicals Mediated Redox Signaling in Plant Stress Tolerance"

_life, 2023, doi:10.3390/life13010204_

Round 1
Reviewer 1 Report
The review may help researchers to understand the stress mechanisms of plants, collaborating with the literature.
Author Response
The authors are highly thankful to the learned reviewer for positive recommendation of our work.
Reviewer 2 Report
Overall review is timely and well written. However, in this 1-2 figures showing mechanism of ROS/RNS/RSS formation, their role and cross talk is missing. It will complete the review.
Quality of Figure 1 and 2 is not up to the mark.
Author Response
Comment 1-
Overall review is timely and well written. However, in this 1-2 figures showing mechanism of ROS/RNS/RSS formation, their role and cross talk is missing. It will complete the review.
Answer 1-. The authors are in complete agreement with the suggestion of learned reviewer regarding the illustration describing crosstalk of ROS-RNS-RSS and the same has been incorporated as figure 3 in section 6 at page no 21.
Comment 2-
Quality of Figure 1 and 2 is not up to the mark.
Answer 2-. The quality of figure 1 and 2 has been improved to 1800 dpi.
Reviewer 3 Report
The manuscript by Krishna Kumar Rai and Prashant Kaushik is about free radicals and their involvement in signaling in plant stress tolerance. According to authors this review describes knowledge about generation of reactive oxygen species (ROS), reactive nitrogen species (RNS), and reactive sulfur species (RSS). Moreover, we can learn how ROS/RNS/RSS modulate cell signaling and gene regulation for abiotic stress management in crop plants.
In my opinion, the whole structure of this work is not very clear. Overall, it's pretty confusing to me. Sometimes the authors write about three types of radicals and sometimes only one. Chapter 6, for example, is mainly about NO. In my opinion, a number of information contained in the text could be shown in diagrams. They are illustrative and often make it easier for the reader to understand, for example, since these radicals often interpenetrate and participate in many signaling pathways, it should be shown graphically.
Remarks:
1. Figures 1 and 2 are too small, you can't see the inscriptions at all.
2. Tables should be formatted differently. They should be slightly reduced.
3. References should be formatted more precisely because there are gaps every few positions as if they were to be paragraphs.
4. Line #338 - It says in the text (Fang et al., 2021 [47]). This should be improved.
5. Reference numbers in the text are written in bold and some in normal. How should it be correct?
In my opinion, this manuscript can be recommended for a publication after taking into account the above doubts.
Author Response
Comment 1-
In my opinion, the whole structure of this work is not very clear. Overall, it's pretty confusing to me. Sometimes the authors write about three types of radicals and sometimes only one. Chapter 6, for example, is mainly about NO. In my opinion, a number of information contained in the text could be shown in diagrams. They are illustrative and often make it easier for the reader to understand, for example, since these radicals often interpenetrate and participate in many signaling pathways, it should be shown graphically.
Answer 1- The authors are in complete agreement with the suggestion of learned reviewer regarding the illustration describing crosstalk of ROS-RNS-RSS and the same has been incorporated as figure 3 in section 6 at page no 21.
Comment 2-
Figures 1 and 2 are too small, you can't see the inscriptions at all.
Answer 2- The quality of figure 1 and 2 has been improved to 1800 dpi.
Comment 3-
Tables should be formatted differently. They should be slightly reduced.
Answer 3-. As per the suggestion of learned reviewer the table 2 has been reformatted and with reduced contents.
Comment 4-
References should be formatted more precisely because there are gaps every few positions as if they were to be paragraphs.
Answer 4-. Corrected as per suggestion
Comment 5-
Line #338 - It says in the text (Fang et al., 2021 [47]). This should be improved.
Answer 5-. Corrected as per suggestion
Comment 6-
Reference numbers in the text are written in bold and some in normal. How should it be correct?
Answer 6-. Corrected as per suggestion